# Radiomics and artificial intelligence in malignant uterine body cancers: Protocol for a systematic review

**Gloria Ravegnini**[1,2☯], **Martina Ferioli**[3,4☯], **Maria Abbondanza Pantaleo**[4,5], **Alessio G. Morganti**[3,4,5], **Antonio De Leo**[4,5], **Pierandrea De Iaco**[2,5]\*, **Stefania Rizzo**[6,7‡], **Anna Myriam Perrone**[2,5‡]

**1** Department of Pharmacy and Biotechnology, University of Bologna, Bologna, Italy, **2** Division of Oncologic Gynecology, IRCCS-Azienda Ospedaliero-Universitaria di Bologna, Bologna, Italy, **3** Radiation Oncology, IRCCS Azienda Ospedaliero-Universitaria di Bologna, Bologna, Italy, **4** Department of Experimental, Diagnostic and Specialty Medicine, University of Bologna, Bologna, Italy, **5** Centro di Studio e Ricerca delle Neoplasie Ginecologiche (CSR), Department of Medical and Surgical Sciences (DIMEC), University of Bologna, Bologna, Italy, **6** Istituto di Imaging della Svizzera Italiana (IIMSI), Ente Ospedaliero Cantonale (EOC), Lugano, Switzerland, **7** Facoltà di Scienze Biomediche, Università della Svizzera Italiana, Lugano, Switzerland

☯ These authors contributed equally to this work.
‡ SR and AMP are joint senior authors on this work.
\* pierandre.deiaco@unibo.it

**Funding:** The author(s) received no specific funding for this work.

**Competing interests:** The authors have declared that no competing interests exist.

## Abstract

### Introduction

Uterine body cancers (UBC) are represented by endometrial carcinoma (EC) and uterine sarcoma (USa). The clinical management of both is hindered by the complex classification of patients into risk classes. This problem could be simplified through the development of predictive models aimed at treatment tailoring based on tumor and patient characteristics. In this context, radiomics represents a method of extracting quantitative data from images in order to non-invasively acquire tumor biological and genetic information and to predict response to treatments and prognosis. Furthermore, artificial intelligence (AI) methods are an emerging field of translational research, with the aim of managing the amount of data provided by the various -omics, including radiomics, through the process of machine learning, in order to promote precision medicine.

### Objective

The aim of this protocol for systematic review is to provide an overview of radiomics and AI studies on UBCs.

### Methods and analysis

A systematic review will be conducted using PubMed, Scopus, and the Cochrane Library to collect papers analyzing the impact of radiomics and AI on UBCs diagnosis, prognostic classification, and clinical outcomes. The PICO strategy will be used to formulate the research

questions: What is the impact of radiomics and AI on UBCs on diagnosis, prognosis, and clinical results? How could radiomics or AI improve the differential diagnosis between sarcoma and fibroids? Does Radiomics or AI have a predictive role on UBCs response to treatments? Three authors will independently screen articles at title and abstract level based on the eligibility criteria. The risk of bias and quality of the cohort studies, case series, and case reports will be based on the QUADAS 2 quality assessment tools.

## Trial registration

PROSPERO registration number: CRD42021253535.

## Introduction

Uterine body cancers (UBC) are represented by endometrial carcinoma (EC) and uterine sarcoma (USa). EC are the most common female reproductive cancer in high-income countries with a growing incidence recorded in recent years [1]. USa are rare tumors and one of the deadliest gynecological cancers [2].

Clinical management of UBCs is complex and requires an accurate assessment of risk stratification factors in order to plan the correct therapeutic strategy. In UBCs the definition of risk categories is complicated by the stratification of tumors in different stages, different differentiation grades, different histological subtypes, and for EC, different biological subtypes based on molecular characteristics. Unfortunately, the evaluation of most of these parameters is operator-dependent, and therefore subject to possible inaccuracies. Furthermore, the need of introducing several parameters in the risk assessment, each of them associated with a certain risk of error, multiplies the probability of incorrect prognostic stratification.

For example, according to the European Society of Medical Oncology (ESMO), the main prognostic factors of EC are represented by disease stage, histological type, degree of differentiation, and vascular lymphatic invasion. However, the assessment of all these parameters is operator-dependent and poorly reproducible, even among experienced operators. All this leads to a non-negligible risk of incorrect classification of EC patients in relapse and death risk classes.

Moreover, additional risk stratification methods have become available over the last few years, based on the molecular features of TCGA [3, 4]. The TGCA published the first comprehensive genomic characterization of EC in 2013 and identified four subgroups: 1) ultramuted–POLEmut EC, harboring pathogenic mutations in the POLE gene, 2) hypermuted—Mismatch repair deficient (MMRd) EC showing a mismatch repair (MMR) defect or microsatellite instability (MSI), 3) copy-number low—mutated TP35 (p53abn) EC with mutations in TP53 and 4) copy-number high, a group with no specific molecular profile (NMSP).

However, this additional stratification system, while able to provide useful prognostic information, complicates the possibility of assessing the risk for the individual patient in daily clinical practice. In fact, an integrated assessment of traditional risk factors (stage, grading, lymphovascular infiltration) together with genetic risk factors is particularly complex and does not find clear indications from international guidelines. In this context, the introduction of predictive models able to manage all these data thanks to the use of AI methods, would be particularly useful.

Regarding USa, a common issue is the difficult differential diagnosis between USa and its benign counterpart (fibroid) [5]. Pathology examination, especially on the surgical specimen,

is frequently the only method allowing a definitive clarification of diagnostic doubts. However, inappropriate resection techniques, such as intraoperative fragmentation of the tumor (morcellation), can lead to dramatically worse outcomes [6].

To solve these issues, standardized evaluation by AI algorithms able to overcome the human cognitive possibilities seems very attractive [7]. In this regard, radiomics and more generally AI-based analyses represent emerging translational research fields aimed at data mining from images and management of complex datasets to develop predictive models, respectively. Both of them, eventually combined with or including gene expression data, could support evidence-based diagnostic and therapeutic decision, in order to allow personalized and precision medicine in this setting [8].

Based on this background, the purpose of this protocol for systematic review is to provide an overview of radiomics and AI studies on UBCs (EC and USa).

## Methods

### Study registration

The protocol was drafted according to the Preferred Reporting Items for Systematic Reviews and Meta-Analyses project (PRISMA-P), S1 Checklist [9]. The protocol was submitted for registration in the PROSPERO International Prospective Register of Systematic Reviews (CRD42021253535) [10]. The study started on 9$^{th}$ April 2021 and is planned to be completed by the end of October 2021. We expect to complete and publish the analysis by June 2022.

### Patient and public involvement

No patient involved.

### Bibliographic search

The systematic review will be carried out in accordance with the PRISMA Statement principles. PubMed, Scopus, and Cochrane Library databases will be systematically searched for original articles analyzing the role of AI/radiomics on UBCs diagnosis, prognosis, and clinical outcomes. Relevant studies will be selected using the Boolean combination of the following key terms: "((uterine neoplasms[Title/Abstract] OR uterine sarcomas[Title/Abstract] OR uterine fibroids[Title/Abstract] OR endometrial cancer[Title/Abstract]) OR ("Uterine Neoplasms"[Mesh])) AND ((radiomics[Title/Abstract]) OR (("Artificial Intelligence"[Majr]) OR (robotics[Title/Abstract] OR AI[Title/Abstract] OR expert system[Title/Abstract] OR expert systems[Title/Abstract] OR intelligent retrieval[Title/Abstract] OR knowledge engineering [Title/Abstract] OR machine learning[Title/Abstract] OR natural language processing[Title/Abstract]))) Filters: Female. Additionally, the reference list of reviews, meta-analyses, and all original studies will be hand-searched to acquire further relevant studies missed from the initial electronic search. The Mendeley bibliographical software will be used to manage references ensuring a comprehensive survey. The Population, Intervention/Comparator, and Outcome (PICO) [11] strategy will be used to frame the search questions: What is the impact of radiomics and AI on diagnosis, outcomes prediction and clinical results in UBSc? Can radiomics or other AI-based analyses improve differential diagnosis between USa and fibroid? Can AI-based predictive models and radiomics anticipate UBCs response to treatments?

### Eligibility criteria

Eligible studies will include clinical studies, case reports, retrospective and prospective studies, case series, and clinical trials. The following studies will be excluded: preclinical studies,

duplicate data, study protocols, systematic or narrative reviews, meta-analyses, letters-commentaries, editorials, surveys, guidelines and recommendations. Moreover, the following criteria for studies selection will be used, based on the PICO framework:

**Participants.** Studies including women affected by UBCs in which radiomics or AI were used to assess specific features to improve diagnosis, risk stratification and treatments.

**Interventions.** Analysis on clinical, radiological, and imaging data based on radiomics or AI in patients with UBCs.

**Outcome measures.** Impact of radiomics on diagnosis and clinical outcomes (tumor response, local control, distant metastases, pattern of failures, overall survival); evaluation of the used segmentation methods (manual, semi-automatic, or automatic); evaluation of predictive models, based on AI, including either radiomics or other clinical features.

## Selection of studies

After duplicate publications removal, three authors will independently screen the retrieved papers based on their title/abstract/keywords to perform a preliminary selection based on the afore mentioned eligibility criteria. After this first screening, the remaining papers will be evaluated by examining the text in full. All differences arising at this stage will be solved by consensus between the researchers. In case of disagreement a fourth author will be involved in the final decision. Then, from the definitively selected papers data will be collected using a specific form including study design, number of patients, imaging type, AI or radiomics method, study objective, tumor histology and stage, validation group, segmentation type, previous therapies, clinical response evaluation criteria, and other prognostic factors. For the excluded papers, the reason for their removal from the analysis will be reported in the PRISMA flow chart (Fig 1). The final information will be verified by a lead author.

## Data collection and analysis

**Data extraction and management.** We referred to the research selection method in the Cochrane collaboration Network system Evaluator Manual V.5.0. Based on the PRISMA flowchart in Fig 1, three researchers will use the Mendeley reference management software to independently screen, cross-check, and verify the retrieved documents according to the review inclusion and exclusion criteria. In case of differences in the selection, the researchers will negotiate with a fourth author to achieve consensus. We will use Excel 2013 to extract relevant information (S1 Table), including:

1. Clinical research (title of the paper, first author name, year of publication, aim and design of the study, primary cancer, tumor stage, sample size, and median/mean age).

2. Validation cohort (present or not and independent or not from the discovery cohort).

3. Imaging techniques used in the study (ultrasonography, computed tomography, magnetic resonance imaging, positron emission tomography).

4. Segmentation technique.

5. Outcome measures.

**Data synthesis and analysis.** Data from the selected papers will be tabulated based on the studies characteristics. Furthermore, reports will be analyzed separately based on the primary tumor (EC or USa), and then they will be further separated according to study characteristics and aims (diagnosis and/or outcomes prediction). Moreover, three authors will independently

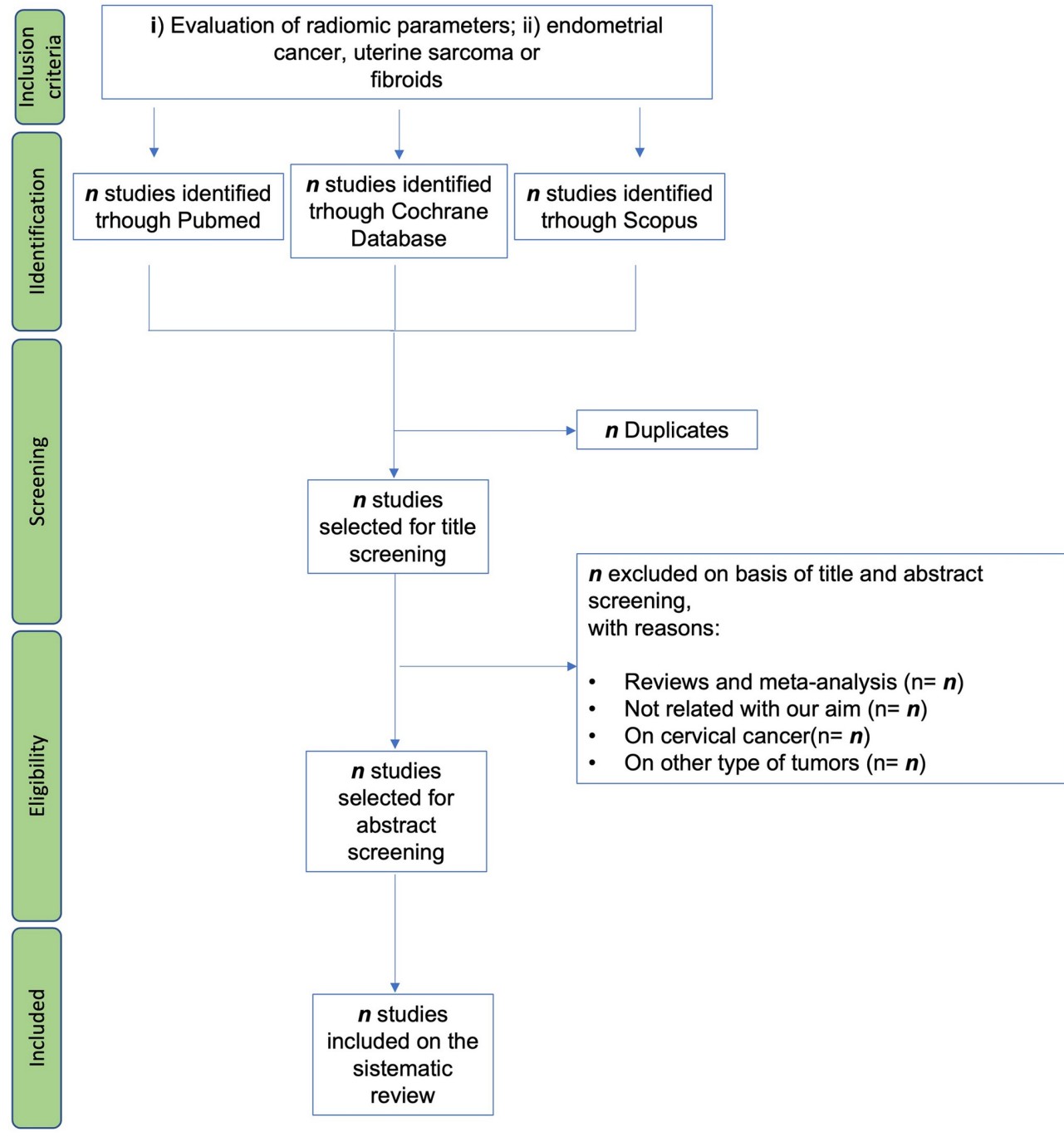

**Fig 1. Shows the studies selection process.**

assess the methodological quality of the selected studies based on the QUADAS 2 quality assessment tools [12]. In case of conflicting evaluations, the final decision was taken by discussion with a fourth reviewer.

Based on the data extracted, the overall quality of the included studies will be critically evaluated based on the QUADAS-2 tool, assessing the four standard domains (patient selection,

index test, reference standard, and flow and timing). Technical aspects of the included studies will be also analysed; in this context, type of techniques (MRI, CT or PET), type of segmentation adopted (Manual, Semi-automatic, Automatic) and how the predictive model was generated (i.e., using radiomics, machine learning, or deep learning. We are also particularly interested in analysing the main purpose of the study (i.e., for staging purpose, for lesion characterization or for survival prediction).

## Discussion

The possibility of non-invasively extract biological and molecular information through images, using radiomics techniques, is obviously of great interest. In fact, the introduction of radiomics analyses in oncology shows growing interest and applications. The radiomics process can be divided into distinct steps such as image acquisition and reconstruction, image segmentation, features extraction and qualification, analysis, and model building [8]. Quantitative image features based on intensity, shape, size or volume, and texture offer information on tumor phenotype and microenvironment (or habitat) that is distinct from that provided by clinical reports, laboratory test results, and genomic or proteomic assays [13]. Furthermore, and more generally, AI is becoming a major player in integrating clinical, pathological, biological, molecular, genetic, and imaging data to develop predictive models [14].

The latter, requiring the use of machine learning or deep learning algorithms [15, 16], will be critical for oncological decision-making in a context where the amount of data available for each patient will be far beyond the human cognitive capacities.

In particular, AI-based analyses of radiological, histological and molecular features could improve the diagnostic and therapeutic pathway of UBCs. In fact, the availability of diagnostic and prognostic biomarkers is still an unmet clinical need in EC and USa [17, 18]. More specifically, integration of multi-omic data through AI systems could bypass the conventional clinical and molecular risk classifications (ESMO risk [19] and TCGA subclasses [3]) to allow a more personalized management of EC. In any case, even the simple use of traditional risk stratification systems requires radiological and pathological evaluations, both of which are subject to possible inaccuracies [20]. The latter can affect the reliability of tumor staging and prognosis prediction, with potential negative effects on clinical outcomes and possible legal disputes. In this context, the use of AI based systems has the theoretical potential to improve the reliability of individual diagnostic methods. For example, radiomics could be the key to solve the complex problem of differential diagnosis between uterine fibroid and USa.

In this scenario, a detailed review of available literature data is needed to summarize and analyze methods and results of AI applications, including radiomics, in the UBCs setting. In particular, the reliability of predictive models based on AI/radiomics will be evaluated based on the presence and type of patient populations used as validation cohorts (internal, external or prospective). Indeed, this point is critical in order to introduce these models in the clinical management of UBCs.

In conclusion, an increasing demand for precision and personalized therapies is required in oncology and particularly in the UBC setting. The aim of this protocol for systematic review is to evaluate all available information about applications of radiomics and AI-based models to improve diagnosis, staging, risk stratification, therapy and follow-up of UBCs.

## Supporting information

**S1 Checklist. PRISMA-P checklist.**
(PDF)

**S1 Table. Excel template used to extract relevant information.**
(XLSX)

## Author Contributions

**Conceptualization:** Alessio G. Morganti, Antonio De Leo, Pierandrea De Iaco, Stefania Rizzo, Anna Myriam Perrone.

**Data curation:** Gloria Ravegnini, Martina Ferioli, Pierandrea De Iaco, Stefania Rizzo, Anna Myriam Perrone.

**Formal analysis:** Gloria Ravegnini, Martina Ferioli, Alessio G. Morganti, Anna Myriam Perrone.

**Supervision:** Pierandrea De Iaco, Stefania Rizzo.

**Writing – original draft:** Gloria Ravegnini, Maria Abbondanza Pantaleo, Alessio G. Morganti, Antonio De Leo, Pierandrea De Iaco, Stefania Rizzo, Anna Myriam Perrone.

**Writing – review & editing:** Gloria Ravegnini, Martina Ferioli, Maria Abbondanza Pantaleo, Alessio G. Morganti, Antonio De Leo, Pierandrea De Iaco, Stefania Rizzo, Anna Myriam Perrone.

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
