## [Decision Letter · Decision Letter 0]

7 Jan 2022

PONE-D-21-29302Radiomics and artificial intelligence in malignant uterine body cancers: protocol for a systematic reviewPLOS ONE

Dear Dr. Perrone,

Thank you for submitting your manuscript to PLOS ONE. After careful consideration, we feel that it has merit but does not fully meet PLOS ONE’s publication criteria as it currently stands. Therefore, we invite you to submit a revised version of the manuscript that addresses the points raised during the review process.

We look forward to receiving your revised manuscript.

Kind regards,

Sandro Pasquali, M.D., Ph.D.

Academic Editor

PLOS ONE

Journal Requirements:

2. PLOS requires an ORCID iD for the corresponding author in Editorial Manager on papers submitted after December 6th, 2016. Please ensure that you have an ORCID iD and that it is validated in Editorial Manager. To do this, go to ‘Update my Information’ (in the upper left-hand corner of the main menu), and click on the Fetch/Validate link next to the ORCID field. This will take you to the ORCID site and allow you to create a new iD or authenticate a pre-existing iD in Editorial Manager. Please see the following video for instructions on linking an ORCID iD to your Editorial Manager account: https://www.youtube.com/watch?v=_xcclfuvtxQ.

Additional Editor Comments (if provided):

Please, carefully address reviewers' comments both in the reply letter and in the manuscript text.

Reviewers' comments:

Reviewer's Responses to Questions

**Comments to the Author**

1. Does the manuscript provide a valid rationale for the proposed study, with clearly identified and justified research questions?

Reviewer #1: Partly

Reviewer #2: Yes

2. Is the protocol technically sound and planned in a manner that will lead to a meaningful outcome and allow testing the stated hypotheses?

Reviewer #1: No

Reviewer #2: Yes

3. Is the methodology feasible and described in sufficient detail to allow the work to be replicable?

Reviewer #1: Yes

Reviewer #2: Yes

4. Have the authors described where all data underlying the findings will be made available when the study is complete?

Reviewer #1: Yes

Reviewer #2: No

5. Is the manuscript presented in an intelligible fashion and written in standard English?

Reviewer #1: No

Reviewer #2: Yes

6. Review Comments to the Author

You may also provide optional suggestions and comments to authors that they might find helpful in planning their study.

Reviewer #1: The Authors describe the protocol they are using to finalize a review on the use of radiomics and artificial intelligence in uterine malignancies.

The project is certainly interesting and very topical but has some weaknesses; in particular, endometrial tumors and uterine sarcomas represent two extremely different pathologies. In the case of endometrial carcinomas the salient point becomes the molecular characterization of the tumor, while in uterine sarcomas the problem is the distinction from fibroids. Linking these two pathologies in the same review doesn't seem like a good idea.

Furthermore, the protocol presented does not show anything innovative compared to a common review, so it is not worthy of separate publication. It will be interesting to see the finished work of the Authors.

The work would also require a careful revision of the english language.

Reviewer #2: Provided as attachment.

7. PLOS authors have the option to publish the peer review history of their article (what does this mean?). If published, this will include your full peer review and any attached files.

Reviewer #1: No

Reviewer #2: No

---

## [Author Response · Author response to Decision Letter 0]

2 Feb 2022

Dear Editor, Thank you for sending me the reviewers' comments and I thank the reviewers for taking the time to make our protocol better. We will answer the questions point by point.

Major Issues

- Add the planned timeline for the study to meet publication criteria

We added the sentence “We expect to complete and publish the analysis by June 2022”

- Add how the data used for the proposed protocol will be made available to meet publication criteria (authors mention information will be extracted using Excel 2013 so that can be possibly be made available).

- Thanks for the comment. We included the excel sheet as supplementary table 2 (line 172)

Minor Issues

- Identification of studies through Scopus seems to be missing in Figure 1

Thanks for the comment, Figure 1 has been modified according to the suggestion

- It will be helpful to have a preliminary analysis as an exemplar of the use of this proposed protocol for a systematic review of radiomics and AI

We added a description of the planned analysis (line 190)

“Based on the data extracted, the overall quality of the included studies will be critically evaluated based on the “Quality Assessment of Diagnostic Accuracy Studies” tool (QUADAS-2), assessing the four standard domains (patient selection, index test, reference standard, and flow and timing). Technical aspects of the included studies will be also analysed; in this context type of techniques (MRI, CT or PET), type of segmentation adopted and how the predictive model was generated (i.e using radiomics, machine learning (ML), or deep learning (DL). We are also particularly interested in analyse the main purpose of the study (i.e for staging purpose or for lesion characterization or for survival prediction).”

- Line 42 (mentions aim of this systematic review), 102 (mentions purpose of this systematic review), 218 (mentions aim of this systematic review) but it technically should be “aim/purpose of this protocol for systematic review” since the paper has not conducted a systematic review yet and is proposing a protocol for it.

Thank you, now Lines 42, 102, 218 have been changed as requested

- Add references to briefly talk about traditional radiomics (such as Gillies et al 2016 and others) which will link to AI radiomics

- Thank you for the comment we added the follows sentences about the radiomics use in line X page Y and the subsequent references are included line 194 page 9: “The radiomics process can be divided into distinct steps such as image acquisition and reconstruction, image segmentation, features extraction and qualification, analysis, and model building [8]. Quantitative image features based on intensity, shape, size or volume, and texture offer information on tumor phenotype and microenvironment (or habitat) that is distinct from that provided by clinical reports, laboratory test results, and genomic or proteomic assays”.

Other Comments

- The proposed protocol can be useful and be applied to perform systematic reviews for other types of cancers.

Thanks for the comment, we definitely hope that our protocol could be useful for other specialty and be cited by other authors

---

## [Decision Letter · Decision Letter 1]

14 Apr 2022

Radiomics and artificial intelligence in malignant uterine body cancers: protocol for a systematic review

PONE-D-21-29302R1

Dear Dr. Perrone,

We’re pleased to inform you that your manuscript has been judged scientifically suitable for publication and will be formally accepted for publication once it meets all outstanding technical requirements.

Kind regards,

Sandro Pasquali, M.D., Ph.D.

Academic Editor

PLOS ONE

Additional Editor Comments (optional):

This study represents a Registered Reports and therefore falls in the scope of PLOS ONE.

Reviewers' comments:

Reviewer's Responses to Questions

**Comments to the Author**

1. Does the manuscript provide a valid rationale for the proposed study, with clearly identified and justified research questions?

Reviewer #2: Yes

2. Is the protocol technically sound and planned in a manner that will lead to a meaningful outcome and allow testing the stated hypotheses?

Reviewer #2: Partly

3. Is the methodology feasible and described in sufficient detail to allow the work to be replicable?

Reviewer #2: Yes

4. Have the authors described where all data underlying the findings will be made available when the study is complete?

Reviewer #2: Yes

5. Is the manuscript presented in an intelligible fashion and written in standard English?

Reviewer #2: Yes

6. Review Comments to the Author

You may also provide optional suggestions and comments to authors that they might find helpful in planning their study.

Reviewer #2: The authors have addressed my earlier comments and I am happy with them although I was expecting some data of papers that have been collected to be present in the excel sheet that was submitted as Supplementary Table 2 instead of just the template. I look forward to reading the systematic review by the authors following this protocol.

7. PLOS authors have the option to publish the peer review history of their article (what does this mean?). If published, this will include your full peer review and any attached files.

Reviewer #2: No

---

## [Editor Report · Acceptance letter]

19 Apr 2022

PONE-D-21-29302R1 

Radiomics and artificial intelligence in malignant uterine body cancers: protocol for a systematic review 

Dear Dr. Perrone:

I'm pleased to inform you that your manuscript has been deemed suitable for publication in PLOS ONE. Congratulations! Your manuscript is now with our production department. 

Kind regards, 

on behalf of

Dr. Sandro Pasquali 

Academic Editor

PLOS ONE